# THE SUPER WEIGHT IN LARGE LANGUAGE MODELS

## ABSTRACT

Recent works have shown a surprising result: a small fraction of Large Language Model (LLM) parameter outliers are disproportionately important to the quality of the model. LLMs contain billions of parameters, so these small fractions, such as 0.01%, translate to hundreds of thousands of parameters. In this work, we present an even more surprising finding: Pruning as few as **a single parameter** can destroy an LLM's ability to generate text – increasing perplexity by 3 orders of magnitude and reducing zero-shot accuracy to guessing. We propose a data-free method for identifying such parameters, termed *super weights*, using a single forward pass through the model. We additionally find that these super weights induce correspondingly rare and large activation outliers, termed *super activations*. When preserved with high precision, super activations can improve simple round-to-nearest quantization to become competitive with state-of-the-art methods. For weight quantization, we similarly find that by preserving the super weight and clipping other weight outliers, round-to-nearest quantization can scale to much larger block sizes than previously considered. To facilitate further research into super weights, we provide an index of super weight coordinates for common, openly available LLMs.

## 1 INTRODUCTION

Large Language Models (LLMs) have been growing in size and capability at an unprecedented rate, enabling them to capture increasingly complex linguistic patterns across a wide range of tasks. However, with this increase in model scale, new and unexpected behaviors have emerged. Dettmers et al. (2022) discovered that once LLMs reach a certain scale, a small set of hidden state features contains outliers of exceptionally large magnitude. These outliers account for a small percentage of all activations but are crucial for preserving the compressed model's quality (Dettmers et al., 2022; Xiao et al., 2023; Wei et al., 2023; Shao et al., 2024).

However, not all outliers are equally important. In this paper, we study a tiny yet important set of outliers in LLMs, termed *super weights*. In Llama-7B, pruning the super weight, a single scalar, completely destroys the model's ability to generate text; the average accuracy of zero-shot downstream tasks effectively plummets to zero. Conversely, pruning the other top 7,000 outliers, including outliers that are larger than the super weight, affects no more than a few percentage points.

Intriguingly, super weights behave similarly across model families and sizes. For one, the super weight is always found in the `mlp.down_proj` weight, always in an early layer. We also find that the super weight amplifies input activation inliers to ultimately produce the exceptionally large magnitude activation observed by Sun et al. (2024) – we term this the *super activation*. This super activation persists throughout the model at exactly the same magnitude and position regardless of the prompt, and we find this is uniquely enabled by skip connections. Finally, super weights suppress stopword likelihood. Taken together, pruning the super weight destroys quality by dampening the super activation and shifting almost all logit probability mass to stopwords.

Both super weights and super activations, which we collectively refer to as *super outliers*, are critical to model quality. Fortunately, there are no more than a handful of scalar super outliers per tensor; in light of this, we revisit round-to-nearest quantization, equipped only with the ability to hold out and restore super outliers. This yields a data-free, hardware-friendly method. For activation quantization, we find this technique competitive with SmoothQuant; for weight quantization, we can scale round-to-nearest to much larger block sizes with higher quality.

**Figure 1: Super Weight Phenemenon.** We discover that pruning a single, special scalar, which we call the *super weight*, can completely destroy a Large Language Model's ability to generate text. On the left, the original Llama-7B, which contains a super weight, produces a reasonable completion. On the right, after pruning the super weight, Llama-7B generates complete gibberish. As we show below, this qualitative observation has quantitative impact too: zero-shot accuracy drops to guessing and perplexity increases by orders of magnitude.

Our contributions are summarized as follows.

1. **Super Weights**: We discover a tiny subset of outliers in LLMs, at most six scalars, that are disproportionately important; pruning these super weights destroys model quality.

2. **Identifying Super Weights**: We present a data-free way to identify super weights using only a single forward pass and provide an index of super weights for existing, open LLMs.

3. **Super Activations**: We analyze how super weights influence inference and relate them to the activation outliers observed in prior work.

4. **Compression**: By preserving super outliers, we show that round-to-nearest quantization increases effectiveness noticeably; preserving super outliers improves compression quality.

## 2 RELATED WORK

### 2.1 OUTLIERS IN LLMs

LLM outliers are widely observed in existing literature. Kovaleva et al. (2021) notes weight outliers, which emerge gradually, beginning early in pre-training, and cause abnormal spikes at select dimensions in the output embedding vectors. Disabling those outliers significantly degrades both the training loss and the downstream task performance. Bondarenko et al. (2021) notes activation outliers, which encourage specific attention patterns, such as attending to the special separator token. However, Sun et al. (2024) first observes an exceptionally extraordinary outlier; in particular, they discover massive activations in LLMs that persist across layers in a fixed position, which Yang et al. (2024) hypothesizes is caused by gated linear units (GLU) and its variants, such as GEGLU and SwiGLU. To mitigate these massive activations, Sun et al. (2024) proposes a learnable attention bias, and (Son et al., 2024; Yang et al., 2024) inserts certain prefixes. To complement these mitigation studies, our focus is instead to leverage, rather than mitigate, these super activations.

### 2.2 OUTLIER-AWARE QUANTIZATION METHODS

Quantization is one of the most popular techniques for reducing LLM resource consumption. However, quantizing LLMs is non-trivial, due to outliers that increase the range of values. Existing works typically study two settings for LLM quantization: (1) Weight-only quantization, where only weights are quantized into low-bit integers; (2) Weight-activation quantization, where both activation and weights are quantized.

For weight-only quantization, several common solutions including using smaller block sizes, to limit the number of values any single outlier can impact (Dettmers et al., 2024; Shao et al., 2024; Dettmers & Zettlemoyer, 2023; Frantar et al., 2022; Dettmers et al., 2023); scaling sensitive weights, via a grid-searched channel-wise scaling, Lin et al. (2024); or clipping outliers via learned optimal thresholds (Shao et al., 2024; Lin et al., 2024). The most common approach is to extract and store sensitive weight outliers in higher-precision (Dettmers et al., 2024; Kim et al., 2024; Dettmers et al., 2022). However, decomposed, mixed-precision arithmetic for hundreds of thousands of weights is unfriendly for hardware and incurs significant latency penalties. We take a different approach, handling at most a half dozen scalars to maintain hardware friendliness.

| Llama-7B | Arc-c | Arc-e | Hella. | Lamb. | PIQA | SciQ | Wino. | AVG | C4 | Wiki-2 |
|---|---|---|---|---|---|---|---|---|---|---|
| Original | 41.81 | 75.29 | 56.93 | 73.51 | 78.67 | 94.60 | 70.01 | 70.11 | 7.08 | 5.67 |
| Prune SW | 19.80 | 39.60 | 30.68 | 0.52 | 59.90 | 39.40 | 56.12 | 35.14 | 763.65 | 1211.11 |
| Prune Non-SW | 41.47 | 74.83 | 56.35 | 69.88 | 78.51 | 94.40 | 69.14 | 69.22 | 7.57 | 6.08 |
| Prune SW, +SA | 26.60 | 54.63 | 56.93 | 12.79 | 67.95 | 61.70 | 70.01 | 50.09 | 476.23 | 720.57 |

**Table 1: Super Weight Importance**. (Section 3) *Prune SW:* Pruning the single, scalar-valued super weight significantly impairs quality – reducing accuracy on zero-shot datasets and increasing perplexity by orders of magnitude. *Prune Non-SW* By contrast, retaining the super weight and instead pruning the other 7,000 largest-magnitude weights marginally affects quality. In other words, a single super weight is more important than even the top 7,000 largest weights *combined*. (Section 3.2) *Prune SW, +SA:* Pruning the super weight but restoring the super activation partially recovers quality. Note that quality is still drastically impaired however, so we conclude that super activations only partially explain how super weights operate. This also shows that super weights and super activations *both* need special handling, to preserve quality.

For activation quantization, there are an increased number of even more aggressive outlier values, making activation quantization more challenging. To tackle this, previous work rotates (Liu et al., 2024; Ashkboos et al., 2024; Chee et al., 2023), clips (Wei et al., 2022) or shifts (Wei et al., 2023; Shao et al., 2024) activations to mitigate activation outliers. One effective approach scales activations (Xiao et al., 2023), migrating the difficulty of quantization from activations to weights with a mathematically equivalent transformation. However, this method – SmoothQuant – requires calibration data to find the optimal hyperparameters. We show a competitive alternative that is alternatively data-free, with a small change to a naive round-to-nearest method.

Recent studies have discovered that activation outliers are associated with weight outliers. The hidden dimensions where activation outliers emerge have a high correlation to sensitive weight channels (Heo et al., 2024; Lee et al., 2024). Along these lines, activation magnitudes have been used as an indicator to find salient weight channels to preserve in weight quantization (Lin et al., 2024). We find the relationship between activations and weights is even more striking: Rather than channel-wise pairs, we find relationships between two individual scalars – up to six weights and one activation.

# 3 SUPER WEIGHTS

Many studies corroborate the importance of weight outliers, showing that a small percentage of the largest magnitude outliers are essential to model quality. This percentage can be as small as 0.01%, but for these billion-parameter models, 0.01% can still include hundreds of thousands of weights. Our investigation reveals a surprising fact about even this group of weight outliers: There exists a single, scalar weight that, despite not being the largest, holds more importance than thousands of other outlier weights combined. We call this single scalar weight a *super weight*.

In our analysis, we find that the super weight is *necessary* for quality, having an outsized influence on quality if removed. Without the super weight, LLMs fail to generate text, resulting in qualitatively (Figure 1) and quantitatively (Table 1) gibberish responses. In particular, zero-shot dataset accuracy is reduced to guessing, and perplexity increases by several orders of magnitude (*Prune SW*). To quantify this influence, we prune all *other* outlier weights (*Prune Non-SW*), comparing the impact of a single super weight against 7,000 other outliers. Remarkably, the accuracy drop associated with pruning this single weight is much greater than the effect of all other outliers *combined*.

## 3.1 IDENTIFICATION OF SUPER WEIGHTS

**Super weights create super activations.** Sun et al. (2024) first discover a handful of exceptionally massive activations, which are crucial to model quality. Massive activations persist across many layers, feature constant magnitude, and always exist at the same position, regardless of input. We find a further intriguing property: The activation's channel aligns with our super weight's, and the activation first appears right after our super weight. To confirm whether this is correlation or causation, we prune the super weight and check the massive activation's magnitude. Per Figure 4, we discover that pruning the super weight drastically reduces the massive activation's magnitude.

| Model | No. | Type | Weight | Coordinates |
|---|---|---|---|---|
| Llama 7B | 2 | mlp | down_proj | [3968, 7003] |
| Llama 13B | 2 | mlp | down_proj | [2231, 2278] |
|  | 2 | mlp | down_proj | [2231, 6939] |
| Llama 30B | 3 | mlp | down_proj | [5633, 12817] |
|  | 3 | mlp | down_proj | [5633, 17439] |
|  | 10 | mlp | down_proj | [5633, 14386] |
| Llama2 7B | 1 | mlp | down_proj | [2533, 7890] |
| Llama2 13B | 3 | mlp | down_proj | [4743, 7678] |
| Mistral-7B v0.1 | 1 | mlp | down_proj | [2070, 7310] |

| Model | No. | Type | Weight | Coordinates |
|---|---|---|---|---|
| OLMo-1B 0724-hf | 1 | mlp | down_proj | [1764, 1710] |
|  | 1 | mlp | down_proj | [1764, 8041] |
| OLMo-7B 0724-hf | 1 | mlp | down_proj | [269, 7467] |
|  | 2 | mlp | down_proj | [269, 8275] |
|  | 7 | mlp | down_proj | [269, 453] |
|  | 24 | mlp | down_proj | [269, 2300] |
| Phi-3 mini-4k-instruct | 2 | mlp | down_proj | [525, 808] |
|  | 2 | mlp | down_proj | [1693, 808] |
|  | 2 | mlp | down_proj | [1113, 808] |
|  | 4 | mlp | down_proj | [525, 2723] |
|  | 4 | mlp | down_proj | [1113, 2723] |
|  | 4 | mlp | down_proj | [1693, 2723] |

**Table 2: Super Weight Directory**. The above layer numbers, layer types, and weight types can be directly applied to Huggingface models. For example, for Llama-7B on Huggingface, access the super weight using `layers[2].mlp.down_proj.weight[3968, 7003]`.

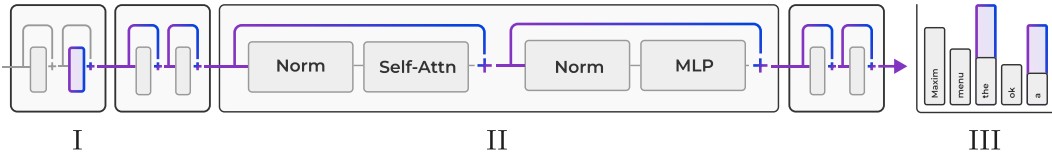

I               II             III

**Figure 2: How Super Weights behave**. *I:* Super weights are often found in an early layer's down projection, indicated with a blue-purple box. The super weight immediately creates an incredibly large-magnitude super activation. *II:* Super activations are propagated through skip connections, indicated with blue-purple lines. *III:* This has a net effect of suppressing stopword likelihoods in the final logits. Removing the super weight causes stopword likelihood skyrocket, indicated with the blue-purple stacked bars. See Appendix A.3.

This suggests that the massive activations are created by super weights. For consistency, we dub these massive activations "*super activations*".

With further investigation, we reveal the mechanism of super weights and super activations. Sun et al. (2024) explained super activations as bias terms, but they did not explain how super activations are created and why they are always in the same positions. Through empirical analysis, we find that before down projection, the Hadamard product of the `gate` and `up` projection creates a relatively large activation, which aligns with the findings of Yang et al. (2024). More importantly, the super weights further amplify it and create super activations.

**Identifying super weight by activation spikes.** Based on the above analysis, we present an efficient way to locate super weights: SWs can be located by detecting the spikes in the `down_proj` inputs and outputs distributions across the layers. This detection only requires a single input prompt, rather than a set of validation data or use-case examples.

Suppose that we have a `down_proj` weight matrix $\mathbf{W} \in \mathbb{R}^{D \times H}$, where $D$ is the dimension of the activation feature and $H$ is the intermediate hidden dimension. Let $\mathbf{X} \in \mathbb{R}^{L \times H}$ be the input matrix, where $L$ is the sequence length. $\mathbf{Y} = \mathbf{X}\mathbf{W}^T$, where $\mathbf{Y}_{ij} = \sum_{k=1}^{d} \mathbf{X}_{ik}\mathbf{W}_{jk}$. Suppose $\mathbf{Y}_{ij}$ is a super activation. If $\mathbf{X}_{ik}$ and $\mathbf{W}_{jk}$ are both outliers that are much larger than other values, $\mathbf{Y}ij$ will be dominated by their product. That is, $\mathbf{Y}_{ij} \approx \mathbf{X}_{ik} \mathbf{W}_{jk}$. In this case, $j$ and $k$ are determined by $\mathbf{X}_{ik}$ and $\mathbf{Y}_{ij}$. Therefore, we start by plotting extreme outliers in the input and output activations of `mlp.down_proj`. Then, we determine the layer and coordinates of the super weight, as illustrated in Figure 3. Once we have detected one super weight, we remove it from the model and repeat the above process, until the magnitudes of large maximum activations are greatly suppressed.

We have identified super weights for commonly available LLMs across different LLM families and model sizes, presented in Table 2. Most of the models we have examined have no more than three super weights. The model with the most super weights, i.e., Phi-3-mini-4k-instruct, contains six. We have also examined the instruction-finetuned models, such as Mistral-7B-Instruct-v0.1 and Llama-2-7B-chat. We find that their super weights are located at the same coordinates as the pre-trained models, which suggests that instruct fine-tuning does not change the position of super weights.

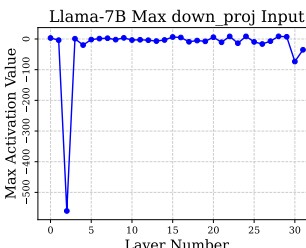 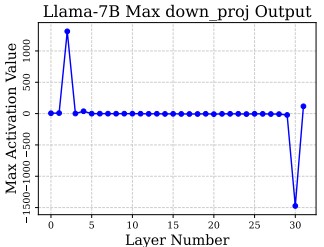 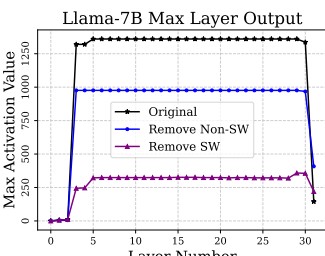

**Figure 3: How to identify the Super Weight** for Llama-7B. `down_proj` input features a large maximum-magnitude activation only in Layer 2, where the super activation first appeared. The value's channel index, e.g., `7003`, tells the row of SW. `down_proj` output likewise features a large maximum-magnitude activation at Layer 2. This value's channel index, e.g., `3968`, gives us the column of the SW.

**Figure 4:** The super activation persists throughout the entire model, at exactly the same magnitude, starting after Layer 2. Pruning the super weight decreases the super activation's magnitude by 75%.

## 3.2 MECHANISMS OF SUPER WEIGHTS

We find that super weights (1) induce super activations, which have lasting effects throughout the entire model, and (2) suppress stopword likelihood (Figure 2).

**Super weights (partially) operate via super activations.** To assess whether the super weight's impact on model quality is solely mediated by the super activations or also by activations of other tokens, we conducted an experiment involving the removal of super weights (SW) and restoration of super activations (SA). Note that a super weight should influence the same channel for all tokens.

We conduct an ablation experiment with three conditions: (1) the original model, (2) remove the super weight (Prune SW), i.e., setting the weight scalar as zero, (3) remove the super weight and restore the super activation at the layer where it first appears (Prune SW,+SA). The third condition allows us to isolate the impact of super weights on super activations only.

Results are shown in Table 1. Specifically, when we restore the super activations, the average accuracy recovers to 49.94 from 35.14, indicating that the restoration of super activations salvaged approximately 42% of the quality loss. These findings suggest that while the super activations contribute substantially to the model's performance, they do not fully account for the super weight's overall influence on quality.

**Super weights affect output token probability distributions.** We studied the impact of super weights with respect to the output token probability distribution, averaged over 500 prompts from Lambaba validation set. We find that when super weights are removed, the stopword probabilities are amplified, e.g., with Llama-7B, the probability of "the" is amplified by around 2×, "." by 5×, and "," by 10× (Figure 5, Appendix Figure 11).

To dive deeper on how SW impact the output token distribution, we conduct a case study with a prompt "Summer is hot. Winter is ". The correct next token should be "cold", which is a word that has strong semantic meaning. With the original model with SW, it correctly predicts the next token "cold" with a high probability 81.4%. However, when the SW is removed, the model's top prediction is a stopword "the" with a non-confident low probability of 9.0%. This indicates that the SW is essential for the model to make a correct and confident prediction of meaningful words.

**Sensitivity of super weights.** We aim to illustrate how variations in the magnitude of super weights impact the model's quality, especially, how does increasing the magnitude affect model quallity. We multiply the super weights by a scaling factor ranging from 0.0 to 3.0. Results in Figure 6 show that amplifying super weights can improve model accuracy, to some extent. See full versions of these plots, for more models and all datasets, in Appendix A.1.

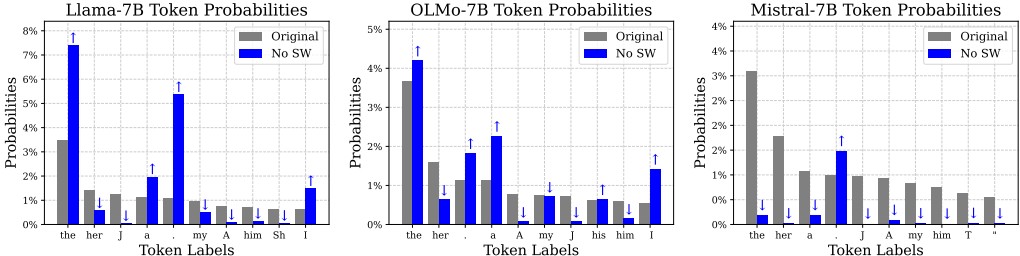

**Figure 5: Super weights suppress stopwords.** Above, we consistently observe that removing super weights results in 2-5× larger stopword probabilities, across a variety of LLMs. At the same time, we observe non-stopwords decrease sharply in probability, reducing by 2-3× to as little as 0.1% probability. Overall, this results in stopwords dominating the highest likelihood tokens.

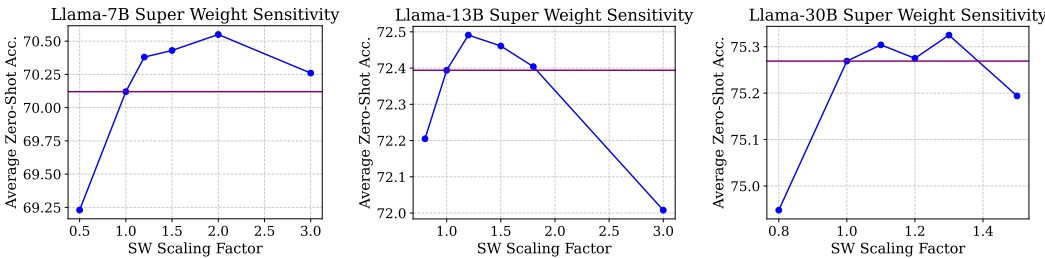

**Figure 6: Amplifying super weight improves quality**. Across model sizes, we consistently observe that there exists *some* scaling where quality is improved. Although the quality improvement is miniscule, a consistent and noticeable trend is surprising, given we're changing only one scalar out of billions. The purple line is the original model's zero-shot average accuracy.

## 4 SUPER-OUTLIER AWARE QUANTIZATION

Quantization is a powerful technique for compressing models and reducing memory requirements. However, the presence of outliers can significantly degrade quantization quality, for both weight quantization and activation quantization. As we mentioned before, we refer to these problematic outliers, both super weights and super activations, as *super outliers*. As we have shown above, these super outliers carry disproportionate importance for model quality, making their preservation during quantization critical. Quantization generally maps continuous values to a finite set of values; we consider one of the simplest forms – namely, asymmetric round-to-nearest quantization:

$$Q(\mathbf{X}) = \text{Round}\left(\frac{\mathbf{X} - \text{MIN}(\mathbf{X})}{\Delta}\right), Q^{-1}(\hat{\mathbf{X}}) = \Delta \cdot \hat{\mathbf{X}} + \text{MIN}(\mathbf{X})$$

where $\Delta = \frac{\text{MAX}(\mathbf{X}) - \text{MIN}(\mathbf{X})}{2^{N-1}-1}$ is the quantization step and $N$ is the number of bits. Note that the maximum value is used to calculate $\Delta$, so super outliers in $X$ drastically increase the step size. With larger step sizes, inliers are rounded to more distant values on average, increasing the quantization error. With increasingly super outliers, inliers are rounded to fewer discrete values, and more quantization bins remain unused. In this way, super outliers cause poor quantization fidelity.

We specifically consider the case where hardware performs arithmetic in half precision, meaning the tensor $X$ is quantized and dequantized before usage; in this setting, we can leverage prior knowledge of super outliers in two ways. First, hold out the super outlier to prevent adverse effects on inlier quantization. Second, restore the super outlier's value after dequantization, to ensure the super outlier's effects are preserved. We adopt this insight in two forms below, for weights and activations.

| **PPL** ($\downarrow$) | Llama-7B | | Llama-13B | | Llama-30B | |
|---|---|---|---|---|---|---|
| | Wiki-2 | C4 | Wiki-2 | C4 | Wiki-2 | C4 |
| FP16 | 5.68 | 7.08 | 5.09 | 6.61 | 4.10 | 5.98 |
| Naive W8A8 | 5.83 *(0%)* | 7.23 *(0%)* | 5.20 *(0%)* | 6.71 *(0%)* | 4.32 *(0%)* | 6.14 *(0%)* |
| SmoothQuant | **5.71** *(100%)* | **7.12** *(100%)* | **5.13** *(100%)* | **6.64** *(100%)* | **4.20** *(100%)* | **6.06** *(100%)* |
| Ours | **5.74** *(75%)* | **7.14** *(82%)* | **5.15** *(71%)* | **6.66** *(71%)* | **4.22** *(83%)* | **6.08** *(75%)* |

**Table 3: Round-to-nearest with super-activation handling is competitive**. *W8A8* is the baseline 8-bit weight and activation quantization, and the small italicized, parenthesized percentages denote what percentage of SmoothQuant's quality improvement is retained. We observe that a naive round-to-nearest, while handling a single scalar super activation per tensor, is competitive with SmoothQuant. Note that SmoothQuant uses calibration data to compute scales, whereas our method does not require data.

### 4.1 ACTIVATION QUANTIZATION

We conduct experiments using round-to-nearest quantization, with a small modification – replace the super activation with the median value (REPLACE), quantize ($Q$) and dequantize ($Q^{-1}$) activations, then restore the super activation in FP16 (RESTORE). This can be expressed as the following,

$$\hat{A} = \text{RESTORE}(Q^{-1}(Q(\text{REPLACE}(A)))) \tag{1}$$

Since the super activation is a single scalar, the bitrate and kernel complexity are not significantly impacted.

### 4.2 WEIGHT QUANTIZATION

Prior art uses (Dettmers et al., 2023; Lin et al., 2024) small group sizes of 64 or 128, as Dettmers & Zettlemoyer (2023) finds that small group sizes are required for precise low-bit quantization. However, the small group sizes come with computational and bitrate overhead, requiring other techniques to handle a high number of half precision scales and biases.

To address this challenge, we propose a simple method to improve INT4 quantization with large blocksizes. First, we identify super weights using Section 3.1. Second, to improve inlier fit, we clip (CLIP) the outlier weights; in this step, the super weight is clipped as well. Quantize ($Q$) and dequantize ($Q^{-1}$) the clipped weights. Then, to ensure the effect of the super weight is preserved, we restore the half-precision super weight after dequantization (RESTORE).

$$\hat{W} = \text{RESTORE}(Q^{-1}(Q(\text{CLIP}_z(W)))) \tag{2}$$

As described in the equation above, we parameterize clipping using a z-score. Assuming all weights fit a Gaussian, we consider all values with a z-score beyond a certain threshold $z$ to be an outlier. To tune this hyperparameter $z$, we find the minimum reconstruction error z-score using 500 examples from the Wikitext-2 train set.

## 5 EXPERIMENTS

To comprehensively demonstrate the effects of super weights, we conduct experiments across LLaMA 7B to 30B, (Touvron et al., 2023), Mistral 7B (Jiang et al., 2023), and OLMo (Groeneveld et al., 2024) [1] To assess the practical application capabilities of LLMs, we evaluate their accuracy on zero-shot benchmarks, including PIQA (Bisk et al., 2020), ARC (Clark et al., 2018), HellaSwag (Zellers et al., 2019), Lambada (Paperno et al., 2016), and Winogrande (Sakaguchi et al., 2021). We use the lm-evaluation-harness (Gao et al., 2024) library to evaluate the above tasks. We also calculate the perplexity for Wikitext-2 (Merity et al., 2017) and C4 (Raffel et al., 2020), following the widely accepted setting from (Frantar et al., 2022).

---

[1]For OLMo, we use their latest Huggingface checkpoints, e.g., `allenai/OLMo-7B-0724-hf`.

| **PPL** ($\downarrow$) | OLMo-1B | | OLMo-7B | | Mistral-7B | |
|---|---|---|---|---|---|---|
| | Wiki-2 | C4 | Wiki-2 | C4 | Wiki-2 | C4 |
| FP16 | 10.12 $_{(100\%)}$ | 12.31 $_{(100\%)}$ | 7.51 $_{(100\%)}$ | 9.52 $_{(100\%)}$ | 5.25 $_{(100\%)}$ | 7.75 $_{(100\%)}$ |
| Naive W8A8 | 10.79 $_{(0\%)}$ | 12.84 $_{(0\%)}$ | 8.70 $_{(0\%)}$ | 10.41 $_{(0\%)}$ | 5.32 $_{(0\%)}$ | 7.83 $_{(0\%)}$ |
| Ours | **10.23** $_{(84\%)}$ | **12.52** $_{(60\%)}$ | **7.80** $_{(76\%)}$ | **9.72** $_{(78\%)}$ | **5.31** $_{(14\%)}$ | **7.81** $_{(25\%)}$ |

**Table 4: Handling the super activation improves activation quantization**. Perplexity $\downarrow$ on Wikitext-2 and C4 for OLMo models and various quantization methods. We can see that simply restoring the scalar-valued super activation after quantizing and dequantizing successfully improves quantization's effectiveness at preserving quality. Notably, note that SmoothQuant does not work on OLMo, as its LayerNorms do not have adjustable parameters. See more results in Appendix A.4

.

## 5.1 ACTIVATION QUANTIZATION

Following SmoothQuant's setting, we simulate W8A8 quantization with FP16 arithmetic. Specifically, we perform 8-bit per-tensor quantization for weights, and 8-bit per-token quantization for activations. We quantize the inputs and weights for linear layers (including q, k, v, gate, up, down projections), and BMM (i.e., batched matmul) in attention layers. For SmoothQuant, we use the default $\alpha$ as 0.85.

We compare our method with SmoothQuant in Table 3. For three Llama models on both datasets, we achieve over 70% of SmoothQuant's improvement over naive quantization. On C4 with Llama-7B and on Wikitext with Llama-30B, we attain above 80% of SmoothQuant's improvement. Our method demonstrates that a significantly simplified approach to quantization can achieve competitive results compared to more complex methods. Unlike SmoothQuant, which applies scales to every activation channel, our method focuses solely on addressing one critical activation outlier.

We extended our evaluation to include additional LLMs: OLMo (1B and 7B), Mistral-7B, and Llama-2-7B. Results are shown in Table 4 and Appendix Table 7. These models represent a diverse set of architectures and training paradigms, allowing us to assess the generalizability of our quantization method. Since SmoothQuant does not report on this set of models, we compare our results with naive W8A8 quantization. Across all models and datasets, our method consistently outperforms naive W8A8 quantization. Our method demonstrates remarkable performance on OLMo models.

Notably, OLMo models use non-parametric LayerNorm, making them incompatible with the SmoothQuant method, which relies on LayerNorm weights to apply the per-channel scales. On Mistral-7B, the improvements are smaller. We hypothesize that this is because the LayerNorm of these models may have learned weights that aggressively suppress the super activation, resulting in a more uniform distribution of activation magnitudes.

These results underscore the critical importance of the super activation in maintaining model performance during quantization. By addressing this single activation with minimal computational overhead, our method captures a significant portion of the benefits achieved by more complex quantization schemes. This finding suggests that the super activation plays a disproportionately large role in preserving model quality during the quantization process.

## 5.2 WEIGHT QUANTIZATION

Recent advancements in LLM quantization techniques have inadvertently highlighted the importance of super weights. Two notable methods, AWQ (Lin et al., 2024) and SqueezeLLM (Kim et al., 2024), demonstrate the significance of preserving these super weights, albeit through different approaches.

### 5.2.1 EXISTING WORKAROUNDS FOR THE SUPER WEIGHT

AWQ, recognizing the need to minimize quantization errors for important weights, introduced a per-channel scaling method. This technique automatically searches for optimal scaling factors, effectively amplifying crucial weight channels. Our analysis of Llama-7B revealed that AWQ scales

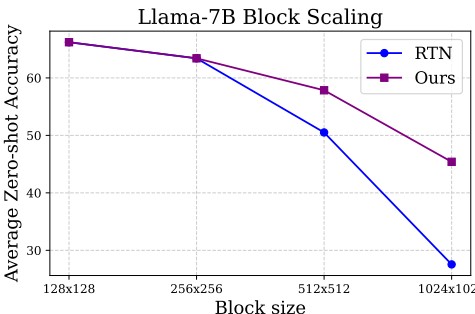 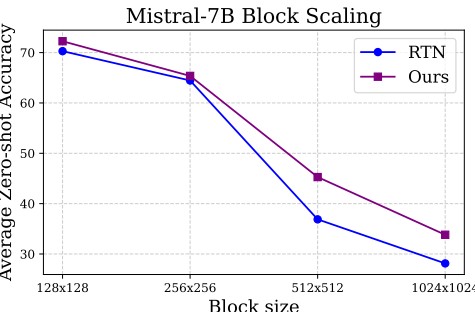

**Figure 7: Restoring super weight improves block scaling**. Smaller block sizes are often used to handle outliers implicitly. We note that block sizes can scale slightly more gracefully by just handling the single scalar-valued super weight.

up the super weight by a factor of 12, corroborating our assessment of the super weight's importance. Similarly, SqueezeLLM proposes a sparse matrix approach that retains the top 0.05% of outlier values in FP16 precision. Our investigation confirmed that this sparse matrix consistently includes the super weights, further validating their importance. Both AWQ and SqueezeLLM, despite employing different strategies, converge on the same principle: protecting super weights is crucial for effective weight quantization in LLMs.

### 5.2.2 SCALING UP BLOCK SIZES

To evaluate the effectiveness of the proposed super weight-aware quantization method, we compare it with the traditional round-to-near quantization approach. We assess the models on a suite of zero-shot downstream tasks, with results illustrated in Figure 7.

In the traditional round-to-near quantization, we observe a clear trend: as the block size increases, model quality significantly degrades. This decline likely results from the increased quantization error introduced when larger blocks of weights are quantized together, which allows outliers to impact more weights. In contrast, our super weight-aware quantization method demonstrates much greater robustness to larger block sizes. As the block size increases, the degradation in model quality is noticeably smaller compared to the round-to-near method. This robustness stems from our method's ability to preserve the most critical weight (the super weight) while minimizing the influence of outlier weights on the overall quantization process. By clipping outliers and focusing on inlier weights, our method maintains higher fidelity in representing the model's parameters.

A key advantage of our method is its ability to support larger block sizes with less loss in model quality. This capability leads to a lower average bitrate and smaller file sizes, which are essential for deploying models in resource-constrained environments, such as mobile devices or edge computing scenarios.

## 6 CONCLUSION

Our study of Large Language Models has revealed the critical role of super outliers – specifically, the super weight and its induced super activation. Although these super outliers are small in number, identifying and preserving them is essential for model quality; pruning the super weight completely destroys the model's ability to generate text, and retaining the super weight can significantly improve the quantized model's quality.

Our findings shed light on how these outliers influence model behavior and provide practical strategies for their detection and management. By sharing a directory of super weights, we furthermore hope to inspire further research into their properties and implications.

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

# A APPENDIX

## A.1 SUPER WEIGHT SENSITIVITY

In this section, we show the full set of results on zero-shot downstream datasets. Results are shown in Table 5 for FP16 models. From the table, we can see that some datasets are more sensitive to super weight (SW) amplification. For example, Winogrande and Lambada show consistent improvements across models when amplifying SW, while PIQA shows same or slightly lower accuracy. We also evaluate the 4-bit quantized Llama-7B with amplified SW in Table 6. We witness similar minor improvements when SW is amplified.

|        | Llama-7B |          | Llama-13B |          | Llama-30B |           | Mistral-7B |          |
|--------|----------|----------|-----------|----------|-----------|-----------|------------|----------|
|        | Original | Amplified | Original | Amplified | Original | Amplified | Original | Amplified |
| ARC-C  | 41.81    | 41.89    | 46.42     | 46.76    | 52.82     | 52.47     | 50.25      | 49.74    |
| ARC-E  | 75.29    | 75.63    | 77.36     | 77.19    | 80.39     | 80.51     | 80.89      | 81.02    |
| Hella. | 56.93    | 56.76    | 59.96     | 59.82    | 63.34     | 63.21     | 61.23      | 61.39    |
| Lamb.  | 73.51    | 74.09    | 76.15     | 76.58    | 77.59     | 77.68     | 75.62      | 75.92    |
| PIQA   | 78.67    | 78.67    | 79.11     | 78.94    | 80.96     | 81.28     | 80.73      | 80.47    |
| SciQ   | 94.60    | 95.40    | 95.00     | 95.30    | 96.10     | 96.10     | 95.90      | 96.00    |
| Wino.  | 70.01    | 71.11    | 72.77     | 72.85    | 75.69     | 76.01     | 73.71      | 74.11    |
| AVG    | 70.12    | 70.51    | 72.39     | 72.49    | 75.27     | 75.33     | 74.05      | 74.09    |

**Table 5:** Accuracy of zero-shot benchmarks of amplifying super weights in FP16 models. The scaling factor is chosen by the highest average accuracy.

|           | RTN-4bit-8x8 |          | RTN-4bit-64x64 |          |
|-----------|--------------|----------|----------------|----------|
|           | Original     | Amplified | Original      | Amplified |
| Llama-7B  | 69.59        | 69.88    | 66.19          | 67.54    |
| Llama-13B | 72.09        | 72.13    | 71.86          | 72.07    |
| Llama-30B | 74.93        | 75.16    | 73.88          | 74.04    |

**Table 6:** Average accuracy of zero-shot benchmarks of amplifying super weights in models with round-to-nearest 4bit weight quantization with blocksizes of 8x8 and 64x64. On quantized models, amplifying super weights also yields a small yet consistent quality improvement.

We visualize the full sensitivity graph starting from zero. We note that the average of all zero-shot datasets is around 30% when datasets are reduced to guessing accuracies.

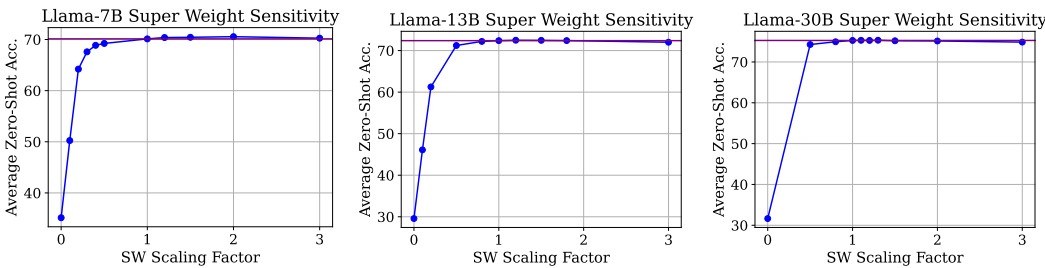

**Figure 8: Amplifying super weight improves quality**. Full results for scaling super weight from 0 to 3.

## A.2 MAXIMUM-MAGNITUDE ACTIVATION OF DOWN PROJECTION MODULE

Below, we show more examples of identifying super weights. We visualize the maximum-magnitude activations in the inputs and outputs of down_proj of all the transformer layers. The outlier "sparks" indicate the row and column index of super weights.

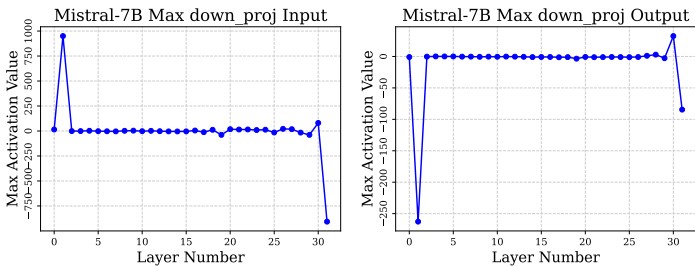

**Figure 9:** Maximum-magnitude activation of `down_proj` across all transformer layers of Mistral-7B.

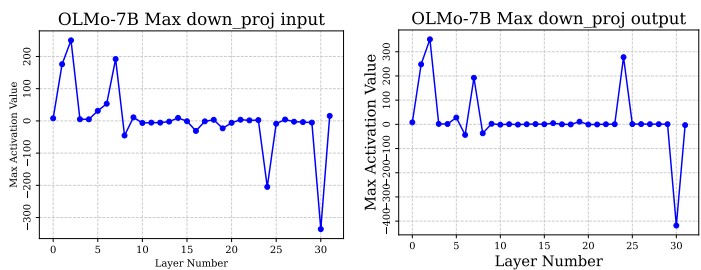

**Figure 10:** Maximum-magnitude activation of `down_proj` across all transformer layers of OLMo-7B.

### A.3   LOGIT DISTRIBUTION WITH SUPER WEIGHT REMOVAL

Below, we visualize more of the logit distribution, when super weights are removed from Llama-7B. Despite the more thorough visualization, the conclusions remain the same: stopwords are amplified and non-stopwords see a drastic decrease in likelihood.

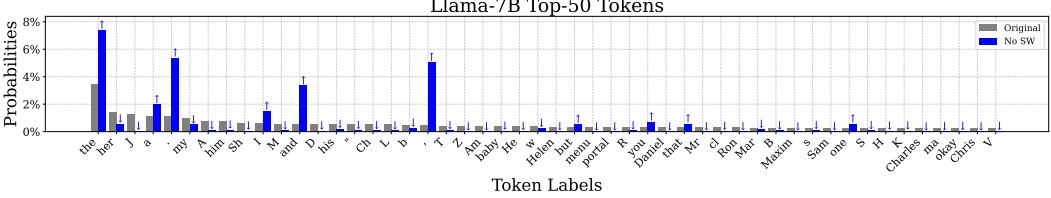

**Figure 11:** Output token distribution before and after removing the super weight on Llama-7B.

### A.4   ADDITIONAL ACTIVATION QUANTIZATION RESULTS WITH SUPER ACTIVATIONS

Below, we include results for Llama-2 7B using our activation quantization. See Table 7.

### A.5   SUPER WEIGHTS AND ATTENTION SINKS

Given that super activations are typically observed on the first token of an input sequence, we hypothesized a potential relationship between super weights and the well-documented phenomenon of attention sinks (Xiao et al., 2024; Son et al., 2024). To test this hypothesis, we conducted experiments comparing attention weight patterns in the presence and absence of super weights. Contrary to our initial expectations, we find that attention sinks persist even when super weights are removed from the model, while not preserving model quality.

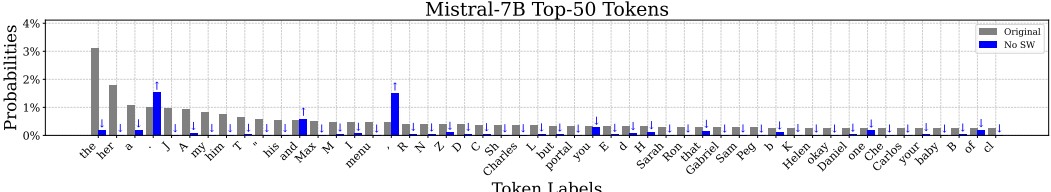

**Figure 12:** Output token distribution before and after removing the super weight on Mistral-7B.

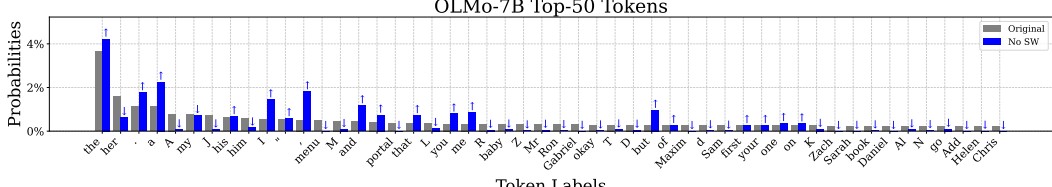

**Figure 13:** Output token distribution before and after removing the super weight on OLMo-7B.

| PPL ($\downarrow$) | Llama-2-7B | |
|---|---|---|
| | Wiki-2 | C4 |
| FP16 | 5.47 *(100%)* | 6.97 *(100%)* |
| W8A8 | 5.58 *(0%)* | 7.09 *(0%)* |
| Ours | **5.57** *(9.1%)* | **7.07** *(16.7%)* |

**Table 7:** Perplexity ($\downarrow$) on Wikitext-2 and C4 with Llama-2-7B.

