# OpenReview forum: "The Super Weight in Large Language Models"
_ICLR.cc/2025/Conference — Submitted to ICLR 2025_

### Official Review · Reviewer_DNuv · 2024-11-01

**Soundness:** 2
**Presentation:** 3
**Contribution:** 2
**Rating:** 6
**Confidence:** 4

**Summary:**

This paper introduces the concept of "super weights" in Large Language Models (LLMs), identifying a small number of individual weight parameters (as few as one) that have a disproportionately large impact on model performance.  Pruning these super weights drastically reduces the quality of generated text, while pruning thousands of other larger-magnitude outliers has a negligible effect.  The paper proposes a data-free method for identifying super weights based on their connection to "super activations," exceptionally large activation outliers previously observed in LLMs.  Finally, the paper demonstrates that preserving super weights and activations during quantization significantly improves compression quality, achieving competitive results methods like SmoothQuant.

**Strengths:**

- The identification of "super weights" and their connection to super activations represents a novel and potentially significant finding in understanding the inner workings of LLMs.
- Connection of "super weights" to quantization accuracy is quite interesting and has practical implications.
- The paper provides a clear methodology for identifying super weights and evaluating their impact, along with an index of super weight coordinates for common LLMs, facilitating further research.

**Weaknesses:**

# Major

- Connection to Adversarial Examples: The literature extensively documents how small changes in the input domain can drastically alter output probabilities. Consequently, significantly harming the network by removing weights, as demonstrated, is somewhat expected.  A discussion addressing the connection between super weight removal and adversarial examples would strengthen the paper.

- Magnitude Pruning Baseline: In Table 1, the comparison of super weight pruning with global magnitude pruning may not be the most informative.  A stronger baseline would involve pruning only within the layer where super activations occur. This would better isolate the impact of the super weight itself.

- Quantization Baseline: The "Naive W8A8" quantization baseline should incorporate clipping. The current presentation makes it unclear whether the observed improvements stem from outlier removal or clipping, especially since super weight handling affects only a single layer during quantization, while clipping is applied to every layer.  Furthermore, it should be noted that the clipping threshold is determined using Wikitext-2, which is also included in the evaluation of quantized models.

# Minor

- Terminology:  The term "extreme" might be more descriptive and informative than "super" when referring to these weights.

- Weight Distribution Visualization: Including a histogram visualizing the position of the super weight within the overall weight distribution would enhance understanding of its magnitude relative to other weights.

**Questions:**

- Section 3.2, "Prune SW+SA":  The description of the "Prune SW+SA" condition in Section 3.2 is unclear.  Specifically, how does this condition differ from the original model?  I understand that super activations typically precede super weights in the model.  Therefore, I am unsure what modification is being made in "Prune SW+SA" and how it distinguishes itself from the original, unpruned model.  Could you please elaborate on this procedure?

---

> ### Author Response · Authors · 2024-11-22
>
> Thank you for highlighting our novel findings on super weights and their practical implications for LLM quantization - we're particularly glad you found the methodology clear and appreciate your recognition of how this work could benefit future research in the field.
>
> ### 1. Connection to Adversarial Examples:
> This is a good point. We agree that there is a similarity between super weights and adversarial examples, however, there is also a fundamental difference between them. We will add the following discussion to the manuscript:
>
> The connection between Super Weights and adversarial examples lies in how both phenomena demonstrate neural networks' sensitivity to small changes. In adversarial examples, minor input perturbations can drastically alter model outputs. Similarly, modifying Super Weights can significantly impact model quality. This sensitivity suggests that neural networks develop highly specialized pathways during training that can be vulnerable to targeted changes.
> However, there is a fundamental distinction between these phenomena. Adversarial examples are input-specific – they exploit vulnerabilities in how the model processes particular inputs. In contrast, Super Weights represent core structural elements whose importance persists across all inputs. While adversarial examples reveal weaknesses in input processing, Super Weights illuminate fundamental aspects of neural network architecture and computation. This input-agnostic nature of Super Weights suggests they represent essential computational primitives that emerge during training, rather than specific vulnerabilities that can be exploited.
>
> ### 2. Magnitude Pruning Baseline
> When we did global magnitude pruning, we set thresholds for each weight tensor to identify outliers, meaning it captures the most significant weights in any given layer already. The fact that removing super weights degrades performance more than removing these globally-identified outliers - including those within the same layer - provides even stronger evidence for their unique importance.
>
> ### 3. Quantization Baseline with Clipping
> We would like to clarify that for activation quantization, we did not apply clipping in both baseline and proposed methods. Are you suggesting incorporating clipping in the RTN baseline in weight quantization? We will be happy to provide the enhanced baseline results, if that's the case.
>
> We would also like to clarify that we use samples from the train set from Wikitext-2 to determine the clipping threshold, while evaluating the models on the test set, which are independent of the train set.
>
> ### 4. Clarification on Prune SW,+SA setting.
> Let's take Llama-7B as an example. In the mlp.down_proj layer, the super weight is the [3968, 7003] element in weight matrix W [4096, 11008]. For a 10-token input, the hidden states X [10, 11008] transform through down_proj to X' = XW^T [10, 4096]. The super weight affects the 3968th channel across all tokens, while a super activation typically appears on just one token.
> The "Prune SW+SA" condition removes both the super weight and any associated super activations to investigate whether SW's impact on model quality operates:
> - Primarily through the single-token SA
> - Also through broader effects across other input tokens

---

> > ### Comment · Reviewer_DNuv · 2024-12-01
> > **Repsonse**
> >
> > I've read authors response. Firstly, I'm sorry about the delay. My question is answered and experiment in Section 3.2 makes sense. However only ~2/3 of my concerns are addressed.
> >
> > 1) input space is, in a way, just another activation. I don't think the difference between adverserial images and activation outliers are as big as it may seem. I think authors should make this connection clear in their work and cite relevant work. Is there, for example, a paper which looks in to effect of "weights" on creating adversarial examples. I think there are few which shows sparse networks are more robust.
> >
> > 2) There are 2 things one do when identifying the super-weight if I recall correctly (1) finding the layer with activation jump (2) identification of super weight by looking at input/output. What I am saying is one can do (1) for the baseline and only do magnitude pruning in corresponding layer. This would show it is not the least magnitude weight. Not you don't search for super weight globally, so it is unfair for the baseline to do that in a away. This could have been done during rebuttal easily, but I guess there was a miscommunication.
> >
> > 3) yes for the weights. Please do. No one really does RTN for quantization. Clipping is a common technique.
> >
> > I will increase my score but I'm still in the middle for this work. It think it is interesting, but some key baselines are missing.

---

### Official Review · Reviewer_72Gg · 2024-11-03

**Soundness:** 3
**Presentation:** 3
**Contribution:** 2
**Rating:** 5
**Confidence:** 4

**Summary:**

This paper reveals that Large Language Models (LLMs) contain a very small subset of weights
(super weights) that are extremely important, where removing them severely degrades model
performance. The researchers developed an efficient, data-free method to identify these super
weights using only a single forward pass. They further investigated how these super weights
influence network behavior by analyzing their relationship with activation outliers. Building on
these insights, they proposed a quantization approach that carefully preserves these super
weights while effectively compressing other weights, resulting in the maintenance of model
quality after compression.

**Strengths:**

The discovery is interesting and the proposed quantization method is easy to implement, which
can maintain better performance compared to Round to nearest quantization with the same
block size.

**Weaknesses:**

The authors failed to show how the proposed methods can improve the SOTA.
1. Although the method is data-free, its performance does not exceed SOTA methods like
SmoothQuant, given incorporating a small calibration dataset would not increase the
quantization complexity much.
2. The author mentions that this method is hardware-friendly, but no experiments to show
its effectiveness in improving latency, throughput, memory usage, etc.

**Questions:**

1. For equation 1, the median is used to replace super activation. Is getting the median
time-consuming since GPU is not good at sorting? (Although there are GPU-version
sorting algorithms)
2. The authors mentioned that SmoothQuant does not report on some models this paper
evaluates, they compare our results with naive W8A8 quantization (line 407 - line 409).
Can the authors run SmoothQuant on these methods since it is open-source? The naive
W8A8 is a too-weak baseline.

---

> ### Author Response · Authors · 2024-11-22
>
> Thank you for acknowledging the discovery of super weights and super activations.
>
> ### W1. Failed to improve SOTA
> Our study's primary objective is not to outperform SOTA methods, but rather to explain why they work effectively.
> Our key finding is that super activation handling appears to be a critical mechanism underlying SOTA methods' success. By achieving comparable performance with a simpler method focused solely on super activations, we demonstrate that while SOTA methods effectively handle these activations, they may simultaneously be processing many unnecessary outliers.
> This insight has important practical implications. It suggests that quantization methods could potentially be simplified while maintaining effectiveness by focusing specifically on super activations rather than broad outlier management. This aligns with a key goal in LLM quantization research: handling as few outliers as possible while maintaining model quality.
> While our method does not exceed SOTA performance, we believe our work makes a valuable scientific contribution by revealing a fundamental mechanism behind SOTA methods' success and demonstrating it through a minimal working example. These insights provide clear design principles for developing simpler, more efficient quantization approaches in future research.
>
> ### W2: Hardware experiments
> In this work, our primary focus was demonstrating the impact of super outliers on quantization. We agree that hardware performance metrics would be valuable in practical applications (but such measurements depend strongly on the hardware chosen), and such analysis does not impact the key observation of this paper: that super outliers exist.
>
> ### Q1: For equation 1, the median is used to replace super activation. Is getting the median time-consuming since GPU is not good at sorting?
> We initially used the median value as a simple placeholder, since  the value will be immediately replaced before computation with the actual super outlier. A better choice, would be to use any inlier value, since again, this is a simple placeholder.  To optimize computational efficiency, we updated our manuscript to specify that instead of a median value, we can use the first element of the tensor, as this requires O(1) time complexity compared to O(n log n) for median calculation. This modification will not affect the final results while significantly reducing computational overhead. We will update the manuscript to reflect this more efficient approach and clarify the rationale behind the placeholder selection.
>
> ### Q2: Run SmoothQuant on more models (line 407 - line 409).
> We will add these comparisons to the final manuscript.

---

> > ### Comment · Reviewer_72Gg · 2024-11-26
> >
> > Thanks to the authors for their response. However, my primary concerns regarding performance improvement over SOTA and hardware efficiency remain unaddressed. I still believe that to make the paper accepted, there should be an improvement either in the perplexity or hardware efficiency to show the value of the findings. Therefore, I will keep my score unchanged.

---

> > > ### Author Response · Authors · 2024-11-27
> > >
> > > > there should be an improvement either in the perplexity or hardware efficiency to show the value of the findings.
> > >
> > > We respectfully disagree that the main value of research is strictly on obtaining SOTA on benchmarks. We ask the reviewer to reconsider their stance when reviewing papers (even if not for this paper) that the main value of research is through SOTA benchmarks: this reviewing strategy discourages publishing an intriguing and previously unreported phenomenon (e.g. that even a *single* weight alteration destroys a multibillion parameter neural network).
> > >
> > > The ICLR reviewing guidelines share this stance when reviewing a paper, asking reviewers the following https://iclr.cc/Conferences/2025/ReviewerGuide
> > > > What is the significance of the work? Does it contribute new knowledge and sufficient value to the community? Note, this does not necessarily require state-of-the-art results.
> > >
> > > This work does contribute new knowledge and sufficient value to our community even though it does not necessarily present state-of-the-art results.

---

### Official Review · Reviewer_rhtc · 2024-11-03

**Soundness:** 1
**Presentation:** 1
**Contribution:** 1
**Rating:** 1
**Confidence:** 5

**Summary:**

This paper investigates the sensitivity of a subset of outliers in LLMs, referring to them as "super weights." The authors conducted experiments to examine the impact of these super weights on model performance.

**Strengths:**

The authors conducted experimental explorations on the so-called "super weights."

**Weaknesses:**

1. The necessity of "super weights" is unclear, as outliers are already identified based on the threshold. Increasing the threshold will naturally reduce the number of outliers with very large weights. Given the known importance of outliers in LLMs, emphasizing "super weights" (outliers at a higher threshold) does not appear novel.

2. Figure 1 is misleading. According to the author's definition, "super weights" are a subset of outliers. However, the figure suggests -1.9 is a typical outlier with nearby values being quite small (.1 and .2), implying that zeroing out outliers produces nonsensical text—a widely acknowledged fact. To better demonstrate the significance of super weights, it would be beneficial to explore whether zeroing out all outliers results in poor performance, and similarly, whether zeroing out just a small subset (e.g., 20-30) leads to comparably severe degradation.

3. Table 1 raises critical concerns. First, the criterion for selecting outliers needs specification. Second, the "Prune SW, +SA" setting in Lines 146-152 is confusing, as it suggests pruning super weights while partially restoring super activations enhances quality. However, the authors did not prune activations, leading to confusion about this claim.

4. Table 2 appears redundant and fails to convey meaningful information. Replacing it with visual representations of "super weights" distributions would be more informative, as the current table occupies considerable space without offering clear insights.

5. Figure 2 is difficult to interpret. The depiction of super weights and their impact, such as generating nonsensical text, is not clear. The use of the same color block in both the network and the output is puzzling. Are the model's dynamics linear? How do the output and weights share the same significance? Clarification is needed on whether this figure is based on assumptions or empirical data.

6. In Lines 189-190, the term "super activations" is introduced but lacks clarity on whether it is threshold-based or aligns with corresponding weights, which could be time-consuming. The authors should clarify this terminology.

7. The paper contains several unprofessional notations. For example, "Yij" should be corrected to "Y_{ij}" in Line 204, and similarly, "Xik" and "Wjk" should be "X_{ik}" and "W_{jk}" in Line 205. The inconsistency in notation and dimensions between "d" and "D" in Line 204 suggests a lack of careful writing and review, raising concerns about the overall professionalism of the paper.

8. Lines 198-210, which discuss the identification of super weights, are crucial yet unclear. The selection criteria for super weights remain ambiguous and need a precise mathematical description. Readers should understand the definition of outliers and the criteria for their selection explicitly.

9. The paper lacks consistency in terminology. "Super weights" sometimes refer to both activations and weights, and at other times only to weights, adding confusion. In Line 306, the term "super outliers" is introduced, suggesting that the paper should maintain consistent terminology from the start, including in the title, if both weights and activations are discussed.

After several careful readings, there are numerous additional concerns throughout the paper. The issues are substantial and critical, making it unlikely to meet the standards of ICLR. I recommend a strong reject based on the quality of this paper and will not change my rate.

**Questions:**

Please refer to the weaknesses section above.

---

> ### Author Response · Authors · 2024-11-22
>
> ### 1. Regarding the necessity of "super weights":
> Our research demonstrates that there is a crucial distinction between general outliers and what we term "super weights." Our key finding is that not all outliers carry equal importance, and their importance is not determined by magnitude. Specifically, we show that super weights, despite not being the largest weights in the network, have disproportionate impact compared to the other 7,000+ outliers combined. This novel observation challenges the conventional understanding of outlier importance in LLMs.
>
>
> ### 2. Regarding the "Prune SW, +SA" setting:
> Our experiments show that pruning super weights (SW) naturally leads to a significant decrease in super activation (SA) magnitudes, shown in Figure 4. The "Prune SW, +SA" experiment was specifically designed to investigate whether super weights' influence is limited to their direct effect on super activations (on a single token) or if they also impact other tokens through different pathways. We have updated the manuscript to make this distinction clearer.
>
> ### Regarding Figure 2:
> The figure illustrates the propagation mechanism of super weights' influence through the network. Specifically, it shows how a super weight in an early layer generates super activation, which then propagates through skip connections to subsequent layers, ultimately affecting token probabilities.
>
> ### Others
> Minor points on terminology and notation, we have updated the manuscript to:
> - Standardize our mathematical notation (e.g., Y_{ij}, X_{ik}, W_{jk})
> - Enhance our figures and tables to better convey our findings
>
> Regarding your statement “I recommend a strong reject based on the quality of this paper and will not change my rate.” We find such firm-willed sentiment in a scientific review panel concerning, and we encourage the reviewer to read the ICLR reviewing guidelines
> “If you believe that a paper has flaws in terms of its evaluation or validation, proofs, or other parts of the discussion, it is critical to point this out to the authors. The authors will have an opportunity to address these concerns, and the iterative process of improving papers after reviewer feedback is important for ensuring the highest quality of ICLR papers.”
>
> This reviewing process is designed to be iterative, where both reviewers and authors have the opportunity to discuss their questions, concerns, and suggestions for improvement.

---

> > ### Comment · Reviewer_rhtc · 2024-11-25
> >
> > Thank you for your response. Unfortunately, I believe the paper has critical issues that render it not yet suitable for acceptance at ICLR. Many of these concerns were outlined in my initial review, which highlighted significant weaknesses across multiple aspects of the work.
> >
> > I have carefully reviewed your response multiple times. However, it did not directly address any key questions or resolve the primary concerns raised in my review. The fundamental issues span across the setting, problem formulation, questions posed, and the methodology, among other aspects, as detailed in my comments.
> >
> > I rarely provide such a strong rating, but in this case, I find it difficult to see the value in the current form of the paper. For these reasons, I must maintain my current rating.

---

> ### Public Comment · ~Yuzong_Chen1 · 2024-11-30
> **Potential misleading reviews generated by LLM**
>
> I appreciate the authors' effort in writing a good paper and Reviewer rhtc for providing the review. However, it seems that this review doesn't point out the real weakness of the paper (most Weakness points are about minor paper writing issues which have been addressed by the authors). The last paragraph of the review, "after several readings" are confusing since the reviewer mentioned additional concerns but didn't list them out. The text structure is potentially obtained from LLMs like ChatGPT. I suggest the area chair to look into this review to provide a more fair assessment of the paper.

---

> > ### Comment · Reviewer_rhtc · 2024-11-30
> > **Responding to this public comment**
> >
> > - However, it seems that this review doesn't point out the real weakness of the paper (most Weakness points are about minor paper writing issues which have been addressed by the authors).
> >
> > This assessment is completely inaccurate. The main problem is not mere grammatical errors, but a fundamental lack of clarity, inadequate explanations, and even contradictory definitions, charts, and explanations. Those things require very substantial improvement, even rethinking the motivation and the foundational assumptions of this work. I believe my review provided detailed evidence with specific examples and data to substantiate each critique.
> >
> > - The last paragraph of the review, "after several readings" are confusing since the reviewer mentioned additional concerns but didn't list them out.
> >
> > The phrase "after several careful readings" was used because I have already identified 9 significant issues in my first few readings, and additional problems persist throughout the paper. However, continuing to list each one would not alter the fact that the paper's overall quality falls significantly below the expected standard. Crucial concerns, such as the protocol for identifying outliers and the subsequent application of super weights, are not discussed at all. This should have been addressed at the outset. Throughout the paper, the term 'weights' seems to have three different meanings: weights, activations, and a combination of weights and activations. This level of confusion extends far beyond mere "minor writing issues."
> >
> > As for the comment on the text structure being derived from LLMs like ChatGPT, I clarify that while ChatGPT was used to check for grammatical consistency, the content of the reviews was written independently by me.
> >
> > Since this is a public discussion, I expect future critiques to be based on substantial, measurable claims. I am not inclined to continue this conversation otherwise.

---

### Official Review · Reviewer_agim · 2024-11-04

**Soundness:** 3
**Presentation:** 3
**Contribution:** 2
**Rating:** 5
**Confidence:** 3

**Summary:**

The paper is about the discovery of super weights in LLMs that are disproportionately important, pruning these hurts model quality quite a bit. The authors have provided a way to identify these super weights using a forward pass. Super weights are activations are sensitive to quantization effects and hence authors propose a super weight aware quantization method enabling effective quantization.

**Strengths:**

Novel discovery about the importance of a few handful of neurons: The identification and analysis of super weights and super activations as critical outliers and their positive influence on model's performance is noteworthy and interesting.

Quantization proposals: Authors went one step further to propose a super weight-aware quantization method to make the best use of these super weights/activations. Data free quantization proposal with on par performance compared to SmoothQuant is also a worthy contribution.

**Weaknesses:**

Though the discovery is quite interesting, the improvements of proposed methods with existing baselines are quite marginal. In general, such kind of super weights might be a natural phenomenon in any machine learning model. How can one say this is relevant only to LLM's?

The work seems to be very much based on empirical observations (which is not my concern) but more discussions/intuitions/explanations around how/why these super weights are formed will be useful.

**Questions:**

The paper mostly focuses on post training model weight/activation analysis and identifies certain handful of importance weights/activations. The authors also say that irrespective of the input prompt the super weights are always the same and they mostly occur in the early layer's down projection with some reasoning via skip connections diagram.

Though these insights are helpful, but it would be good if authors can follow up with what happens during the training process that such super weights are formed in the first place. Does the training methodology in terms of quantization during training/layernorm, gradient scaling, etc play any role in the forming of these super weights?

---

> ### Author Response · Authors · 2024-11-22
>
> Thank your for acknowledging the importance of the discovery of super weights.
>
> ### 1. The improvements of proposed methods with existing baselines are quite marginal.
> We would like to clarify that our study's primary objective is not to outperform SOTA methods, but rather to explain why they work effectively.
>
> Our key finding is that super activation handling appears to be a critical mechanism underlying SOTA methods' success. By achieving comparable performance with a simpler method focused solely on super activations, we demonstrate that while SOTA methods effectively handle these activations, they may simultaneously be processing many unnecessary outliers.
>
> This insight has important practical implications and is of broad interest to the research community. It suggests that quantization methods could potentially be simplified while maintaining effectiveness by focusing specifically on super activations rather than imprecise outlier management. This aligns with a key goal in LLM quantization research: *handling as few outliers as possible while maintaining model quality*.
>
> While our method does not exceed SOTA performance, we believe our work makes a valuable scientific contribution by revealing a fundamental mechanism behind SOTA methods' success and demonstrating it through a minimal working example. These insights provide clear design principles for developing simpler, more efficient quantization approaches in future research.
>
> ### 2. How super weights are formed during training.
> A comprehensive analysis of training dynamics is planned for future work; we can share some preliminary findings based on our analysis of OLMo-1b training checkpoints.
> We traced the evolution of super weight magnitudes across training steps (see plots [here](https://imgur.com/a/fwfdN45)), with each curve representing an individual super weight. Our observations revealed two distinct phases that we will further study in future work:
> 1. Underfitting Phase (0-100k steps):
> - Super weights exhibit rapid magnitude growth
> - Their magnitudes surpass other outlier weights significantly
> 2. Overfitting Phase (100k-700k steps):
> - Super weight magnitudes gradually decrease
> - This decline is likely attributable to the weight decay mechanism

---

> ### Author Response · Authors · 2024-12-02
>
> Dear reviewer agim,
>
> Since the rebuttal deadline is approaching, we would like to check in about our response to your review. We would appreciate your thoughts on our clarifications and reconsideration on the rating.
>
> Thanks!

---

> > ### Comment · Reviewer_agim · 2024-12-02
> >
> > Thank you for the responses to my questions. It does sound like answers to my questions are mostly part of planned future work. I have read other reviewers comments and discussions as well and at this point I would like to keep my rating purely because submitting a more comprehensive paper with additional understanding and improvements will make it a stronger contribution to the community in the next iteration.

---

### Official Review · Reviewer_zzW7 · 2024-11-05

**Soundness:** 3
**Presentation:** 3
**Contribution:** 2
**Rating:** 6
**Confidence:** 4

**Summary:**

This paper focuses on the impact of outlier weights in large language models (LLMs), specifically larger weights, which the authors term superweights and superactivations. First, the authors analyze how much these weights and activations affect LLM performance. They then use this as motivation to discuss quantization methods designed to account for superweights and superactivations. Throughout the paper, the authors also discuss the impact of superweight scaling and provide experimental results showing how their quantization method improves upon standard rounding, especially when using larger block sizes within the network.

**Strengths:**

The paper is well-written and effectively illustrates the importance of superweights and superactivations. I appreciate the discussion on the percolation of superactivations across the network and the identification of superweights across layers (Figure 3). Additionally, I find the potential implications of superweight upscaling presented in Figure 6 quite interesting.

**Weaknesses:**

While I appreciate the analysis presented in this paper, I am struggling to see the novelty of this work. I may be misunderstanding, but from what I gather, superweights and superactivations have already been discussed in prior analyses of LLMs. Additionally, it seems that methods like AWQ and SqueezeLLM inherently focus on superactivations. Furthermore, compared to other weight quantization techniques, the proposed method does not appear to offer significant improvements.

**Questions:**

1. Could the authors provide clarification on the points I raised in the weaknesses section, especially if I may have misunderstood some of the contributions?

2. In Figure 6, do the authors have any insights into the concave behavior of the scaling factor? Are there specific explanations or potential methods for identifying this optimal scaling factor?

3. Regarding the stop word shift in distribution, is it generally accepted that a higher probability of stop words negatively impacts LLM performance?

---

> ### Author Response · Authors · 2024-11-22
>
> ### 1. The novelty of discovering super weights and super activations.
> Our work makes two key novel contributions:
> 1) We systematically identify and characterize a tiny set of "super weights" that are crucial for model performance (even a single weight). Prior studies noted weight outliers but did not isolate this essential subset or demonstrate their necessity.
> 2) We establish a causal relationship between super weights and super activations, showing that these extreme activation patterns emerge specifically from super weights. This advances beyond previous work that observed super activations without showing where they stem from.
>
>
> ### 2. Comparison to other quantization methods.
> Our key finding is that super activation handling appears to be a critical mechanism underlying SOTA methods' success. By achieving comparable performance with a simpler method focused solely on super activations, we demonstrate that while SOTA methods effectively handle these activations, they may simultaneously be processing many unnecessary outliers.
> This insight has important practical implications. It suggests that quantization methods could potentially be simplified while maintaining effectiveness by focusing specifically on super activations rather than broad outlier management. This aligns with a key goal in LLM quantization research: handling as few outliers as possible while maintaining model quality.
> We believe our work makes a valuable scientific contribution by revealing a fundamental mechanism behind SOTA methods' success and demonstrating it through a minimal working example. These insights provide clear design principles for developing simpler, more efficient quantization approaches in future research.
>
> ### 3. In Figure 6, do the authors have any insights into the concave behavior of the scaling factor? Are there specific explanations or potential methods for identifying this optimal scaling factor?
> Thank you for this insightful observation about the concave behavior. We believe the scaling curve's shape can be explained by considering the dynamics of model training, particularly in later stages. As pre-training progresses and the language modeling loss becomes small, the weight decay term begins to dominate the loss function. This regularization effect actively encourages weights to maintain smaller magnitudes, potentially preventing super weights from naturally reaching their optimal scale during training.
> This perspective suggests an interesting future direction: using a small calibration dataset to "finetune" just the super weights, allowing them to reach their optimal scale without the constraints imposed by weight decay during pre-training. This targeted approach could help identify more optimal scaling factors while maintaining the model's learned representations. We will update the manuscript to further clarify this point.
>
>
> ### 4. Regarding the stop word shift in distribution, is it generally accepted that a higher probability of stop words negatively impacts LLM performance?
>
> The relationship between stop word probabilities and LLM performance is context-dependent. In tasks like LAMBADA, where the goal is predicting meaningful target words, our evidence suggests that higher stop word probabilities indeed negatively impact LLM performance. This manifests in two ways:
> In next-word prediction tasks like LAMBADA (predicting a passage's final word), increased stop word probability is resulted from the lower probability of meaningful target words across samples.
> Our case study (lines 258-262) demonstrates that removing super weights causes the target word probability to plummet from 81.4% to under 9%, indicating severe degradation of language modeling capability.

---

> > ### Comment · Reviewer_zzW7 · 2024-11-27
> >
> > I thank the authors for their careful response and appreciate their efforts. I can now more clearly see the impact and noverlty of the work and method I believe the results are important for a better understanding of LLM functionality, irrespective of the practical implications on quantization. I have decided to raise my score to a 6.

---

> > ### Public Comment · ~Xi_Wang4 · 2024-12-05
> > **Comment on author response**
> >
> > This is a very interesting paper! But I have a small question about the author's response:
> >
> > > As pre-training progresses and the language modeling loss becomes small, the weight decay term begins to dominate the loss function.
> >
> > Notice that this is true for Adam + L2 regularization, which is rarely used in practice and LLM pre-training. However, I think this is much less likely to be true for AdamW, where an explicit, derivable coefficient for the L2 regularization does not exist.
> >
> > With that said, I still agree with the authors that weight decay could be playing some very important role in super weights related observations!

---

### Meta-Review · Area_Chair_M6eu · 2024-12-19

**Metareview:**

This paper explores the impact of superweights—defined as weights with larger magnitudes—on the performance of large language models (LLMs). The authors analyze the influence of these superweights on LLMs’ performance and propose specialized quantization methods tailored to superweights. The experimental results show some performance improvements achieved by the method, particularly when employing larger block sizes within the network.

However, three reviewers have raised notable concerns regarding the limited experimental improvements, the lack of clarity in the experiments, and also lack of deep analysis. Furthermore, one reviewer has questioned the definition and conceptual clarity of the superweight concept. As a result, the current version of the paper does not appear ready for publication. We recommend that the authors thoroughly address these issues in alignment with the reviewers' feedback.

**Additional Comments On Reviewer Discussion:**

Here I list the main issues:

1.	Challenging the novelty of this work (Reviewer zzW7):
The authors have adequately addressed this concern, leading to a positive score from the reviewer.

2.	Marginal experimental improvement (Reviewers agim and 72Gg):
The authors did not provide additional experiments but explained that their findings are intriguing. However, both reviewers remain unsatisfied with the response.

3.	Unclear experimental explanations (Reviewers zzW7, rhtc, DNuv):
The authors addressed the concerns point by point, resolving many issues raised by the reviewers.

4.	Empirical observations lacking discussion/intuition on superweight formation (Reviewer agim):
The authors presented experimental findings but did not offer in-depth explanations. The reviewer found the response not so sufficient.

5.	Hardware efficiency (Reviewer 72Gg):
The authors did not provide additional experiments, and the reviewer believes the claims of hardware efficiency are overstated.

---

### Decision · Program_Chairs · 2025-01-22

Reject